# An open-source system for efficient clinical trial support: The COMET study experience

**Jonathan Clutton**[1], **Robert Neal Montgomery**[1], **Dinesh Pal Mudaranthakam**[1], **Erin M. Blocker**[2], **Ashley R. Shaw**[1], **Amanda N. Szabo Reed**[1], **Eric D. Vidoni**[1] *

**1** University of Kansas Medical Center, Kansas City, Kansas, United States of America, **2** Emporia State University, Emporia, Kansas, United States of America

* evidoni@kumc.edu

## Abstract

Exercise clinical trials are complex, logistically burdensome, and require a well-coordinated multi-disciplinary approach. Challenges include managing, curating, and reporting on many disparate information sources, while remaining responsive to a variety of stakeholders. The Combined Exercise Trial (COMET, NCT04848038) is a one-year comparison of three exercise modalities delivered in the community. Target enrollment is 280 individuals over 4 years. To support rigorous execution of COMET, the study team has developed a suite of scripts and dashboards to assist study stakeholders in each of their various functions. The result is a highly automated study system that preserves rigor, increases communication, and reduces staff burden. This manuscript describes system considerations and the COMET approach to data management and use, with a goal of encouraging further development and adaptation by other study teams in various fields.

## Introduction

Lifestyle modification approaches such as exercise are promising interventions for improving brain and cognitive health in older adults [1–3]. However, the scientific consensus is unclear on whether these approaches reduce the risk for cognitive decline and dementia [4, 5]. This lack of consensus has been attributed to the quantity and quality of studies [6]. Larger, high-quality randomized control trials (RCTs) are needed [6] and, in fact, underway [7]; however, these large-scale clinical lifestyle modification trials carry their own unique challenges and issues–increasing protocol complexity [7, 8], multiple stakeholders [9], and at risk for decreased data quality [10, 11].

RCTs with high complexity and aggressive enrollment goals are particularly susceptible to decreased data quality [10]. Poor data quality can result from several factors including, but not limited to, insufficient operational support, protocol interpretation errors, documentation errors, and coding errors [12, 13]. In our experience [7, 8, 14–18],exercise and other lifestyle modification approaches in particular are susceptible to these challenges: long intervention periods; many regular intervention points less readily "dose-able" than a discrete pill or infusion; difficulty in measuring exact dosage outside an experimental environment; overlap/interaction of intervention and daily activity, among other challenges, complicate rigorous execution.

**Data Availability Statement:** Emphasizing Open Science and FAIR Data principles [31] we have created a Git repository to share this work freely with the hope that our system will encourage other

**Funding:** This work was funded by grants from the National Institute on Aging P30 AG072973, R01 AG070036, K01 AG072034, and the Leo and Anne Albert Charitable Trust which do not necessarily endorse the statements. The content is solely the responsibility of the authors and does not necessarily represent the official views of the National Institutes of Health or other funders. The funders had no role in study design, data collection and analysis, decision to publish, or preparation of the manuscript.

**Competing interests:** The authors have declared that no competing interests exist.

Modern electronic data capture (EDC) systems and wearable/mobile technologies have been shown to address some of these issues, but only if they are intuitive and highly usable [19]. Poor EDC usability has been linked to staff frustration and inefficiency [20], while improved usability has been shown to increase data quality [21]. Many other strategies have been suggested to address poor data quality including clearly defined protocols, quality assurance, training, performance-monitoring reports, auditing, and retraining [6, 22]. Most of these strategies are time- and expertise-intensive, limited and valuable commodities for complex, large-scale community-based exercise trials running on tight grant budgets.

The University of Kansas Alzheimer's Disease Research Center (KU ADRC) has a long history of successfully executing complex, large-scale community-based exercise trials [7, 8, 14–18]. In addition to data quality and experimental design challenges noted previously; these studies have a few commonly occurring issues–self-reported data, multiple stakeholders, and tension between scope and immediacy [23]. Common stakeholders include participants, trainers, study staff, investigators, Data Safety Monitoring Committees (DSMC), Institutional Review Boards (IRB), and sponsors with each stakeholder needing a unique set of information. Often, systems or external pressures impel study teams to support some stakeholders over others, i.e., federal reporting requirements and formats often take precedent.

Another common reporting challenge is the collection of intervention adherence and performance data, for example, collecting exercise data from participants. Traditionally, we have kept participant exercise logs at community exercise facilities. These paper logs are accessible to intervention personal trainers and participants. However, limited study staff time makes regular site visits and data collection challenging, leading to a time delay in gathering crucial information, like temporary exercise cessation or modification.

In large, community-based, exercise trials, the time and expertise of study staff and investigators are often in tension between the demands of screening and enrolling, in addition to the quality of the intervention. Financial and environmental challenges such as testing space availability or equipment malfunction are also common [6, 22, 24, 25]. In our ongoing randomized controlled single-site trial, the COMbined Exercise Trial (COMET; NCT04848038; R01 AG070036) [26] nearly 3 full-time equivalent personnel are needed for phone screening, intervention oversite, visit administration, and 16 intervention/safety calls, leaving little time for data entry, quality assurance, and aggregate reporting that is required by a modern, NIH-funded, IRB-overseen clinical trial. The scope and immediacy of the many competing demands on the study team and participants requires new approaches to quality assurance, auditing, retraining, etc. Thus, the COMET team has developed an intuitive and highly usable data management system to address several data management, quality assurance, and reporting needs. The system reduces time- and expertise-burden to satisfy multiple stakeholders, while freeing staff time for skilled activities. Here, inspired by other investigations of trial methodologies [27] we detail the development and deployment of the system using freely available technologies, with the hope that it may spur further innovation for lifestyle clinical trial management and infrastructure development.

## Methods

COMET is a 1-year randomized controlled trial seeking to enroll 280 cognitively normal, older adults aged 65 to 80 to test the independent and combined effects of aerobic and resistance training on cognition and brain function. Prospective participants are recruited as a convenience sample from the community through in person events, mailings, and advertisements in local media. Participants are equally randomized (1:1:1:1) into one of four treatment groups: 1) Core and Fusion Training, 2) Aerobic Training, 3) Progressive Resistance Training, and 4)

Combined Resistance and Aerobic Training. Cognitive function was the pre-specified primary outcome. Structural brain changes, physical function and quality of life were ancillary outcomes of interest. Additional measures of functional brain activity change and blood-based biomarkers were identified as exploratory outcomes of interest. Identifiable information is available to the study team for the purposes of communication and safety. Principal investigators are blinded to *a priori* specified primary and secondary outcomes. The University of Kansas Medical Center Institutional Review Board (IRB) reviews and approves the intervention protocol (#146904). Written informed consent is obtained from all participants in the COMET study. The protocol described in this peer-reviewed article is published on protocols. io (https://dx.doi.org/10.17504/protocols.io.rm7vzxo25gx1/v1) and is included for printing purposes as S1 File.

The COMET protocol includes in-person Baseline, Week 26, and Week 52 outcomes testing, 9 check-in calls, daily activity monitoring, weekly participant exercise logging and regular intervention fidelity checks with interventionists. Following a previously published partnership model, the intervention is performed 3–5 times weekly under personal trainer supervision at participating community fitness centers [9]. At the time of publication 14 fitness centers were participating, including 12 in the Kansas City metropolitan area, and two facilitates in two rural Kansas communities, accommodating approximately 100 active exercising participants. Additional, independent exercise sessions are also prescribed. Personal trainers, (n = 59) undergo quarterly fidelity checks and yearly re-trainings. Data collection includes a cognitive battery, quality of life and exercise agency surveys, dual X-ray absorptiometry, blood collection, one-repetition maximum tests, maximal aerobic capacity, MRI, weekly exercise minutes and intensity logs, safety reports, and actigraphy.

COMET required a data management system that could serve multiple stakeholders and support a wide range of activities. Data originate from disparate sources and locations, necessitating a highly coordinated data management system to ensure quality, completeness, and validity. During study startup, the investigative team worked with the biostatistician to identify best practice principles and available resource to support the project. Guiding needs were financial limitations, existing institutional approval for software and data management solutions to minimize delays in implementation, simplicity of development language for hobbyist coders with limited formal training. The team settled on the use of REDCap, R (Cran R, r-project.org) and associated packages approved for use and hosting on a firewall-secured, university server by the institutional information security department. Data access and edit rights are role-based, allowing for blinding as appropriate. R was used as the primary scripting language, with any graphical and human-readable reports produced with Markdown.

Our first step in developing the data management system was to identify the required data sources and collection methods, standard processes for curating, and appropriate accessibility for study team and investigators. As such, the following sections present the data management decision making process regarding the Data Sources, and Curation.

Next, to support stakeholders, we surveyed clinical research professionals in various roles to index common work activities and potential opportunities for deployment or expansion of our data management system. We asked KU ADRC team members to estimate the time requirements of common study staff responsibilities, i.e. participant communication, data and report compilation, error identification, data transfer, etc. Roles include investigators, post docs, research assistants, research coordinators, students, clinicians, nurses, psychometrists, administrative personnel, and other support positions. All staff were asked to complete the survey, but responses were self-selected. Of 101 staff members, 29 completed the survey– 11 Research Coordinators, 5 Investigators, 3 Students, 2 Nurse Practitioners, 2 Research Assistants, 1 Brain Health Navigator, 1 Director of Engagement, 1 Research Nurse, 1 Sr.

Table 1. Time requirements of common study staff responsibilities (in minutes).

| | Min | Q1 | Med | Mean | Q3 | Max | n |
|---|---|---|---|---|---|---|---|
| Time to send email to participant about study activities (per email) | 2 | 5.00 | 10 | 9.38 | 10.0 | 20 | 13 |
| Time to call about upcoming visits, missing data, intervention info, etc. (per call) | 2 | 5.00 | 5 | 9.14 | 10.0 | 30 | 14 |
| Time to send text message to participant (per text) | 1 | 1.00 | 2 | 2.67 | 4.5 | 5 | 6 |
| Time to compile enrollment, demographic, or other data to make decisions about a study (per compilation) | 10 | 10.00 | 15 | 28.00 | 15.0 | 90 | 5 |
| Time to compile CROMS or RPPR data (per compilation) | 5 | 46.25 | 630 | 916.25 | 1,500.0 | 2,400 | 4 |
| Time identifying coordinator-related issues (per week) | 6 | 75.00 | 240 | 339.00 | 345.0 | 1,680 | 14 |
| Time compiling Data and Safety Monitoring Report (per compilation) | 60 | 75.00 | 90 | 110.00 | 135.0 | 180 | 3 |
| Time transferring data from a source doc to and EDC (per week) | 5 | 60.00 | 60 | 146.81 | 120.0 | 900 | 11 |

Administrative Assistant, 1 Assistant Professor, and 1 Statistician. Staff who worked on the COMET study, with similar data management systems, or were new to the center were removed (n = 3). Surveying of staff was approved as a Quality Improvement project by the University of Kansas Medical Center IRB. A brief written summary and statement approved by the IRB regarding voluntary nature of the survey was included with the survey, consistent with guidance and requirements of the IRB. The data collected on staff effort are not necessarily generalizable to the larger workforce. Table 1 summarizes reported time spent in various study activities.

## Results

### Data sources

COMET data arise from several sources. The study team has committed to minimizing the number of human touches to data, emphasizing automated transfer and electronic storage. This approach has many advantages, such as reduced workload, reduced opportunity for entry error, and greater access to full source documentation for future analysis. However, this approach reduces opportunities for staff to identify errors, and requires a more advanced curation system. For COMET, data originate through– 1) direct entry into REDCap, 2) an electronic data file generated during data collection, or 3) the proprietary actigraphy monitor servers that all participants wear (Fitbit Inc.) servers. All non-REDCap, non-Fitbit source documents and reports are first saved to a quarantine directory. An overnight process, scheduled via cronjob, checks for overlap, file type, file naming, and file contents for consistency between the incoming quarantined data, a permanent frozen dataset, and a working data set. If these checks pass, a copy of the new data is added to the permanent frozen data set, a sequestered data directory that is untouched throughout the trial. Another copy is added to the working data directory and is integrated into the project, and the data is removed from the quarantine directory. Frozen data remain the true, untouched record of all generated data.

The primary outcome for COMET is a global measure of cognitive function as measured by a cognitive test battery composite. Component tests were selected from or considered analogous to, tests from the IGNITE study (NCT02875301). Four tests are administered on paper and scores are directly entered into REDCap. Three tests are administered on an iPad (Apple Inc) via the NIH Toolbox app for iOS. Comma separated volumes (CSV) are emailed to the administering psychometrist who transfers them to the quarantine folder. Five tests are administered via ePrime (Psychology Software Tools, pstnet.com) on a workstation running Windows 10. Like with the iPad, output text files are transferred to the quarantine folder by the administering psychometrist.

Ancillary data sources included health and activity surveys directly entered via REDCap by participants using the stock email survey system. Brain structure and function data are stored on an XNAT server in raw DICOM format. Images reconstructed into NIFTI format are checked for appropriate sequence parameters and visual quality using a semi-automated approach and stored in the raw reconstructed format on university servers. The test administrator captures physical function data via direct data entry into REDCap. Dual x-ray absorptiometry (DXA) reports are exported from the Lunar IDXA system (GE Lunar Corp.) as text files and transferred to the quarantine directory by the test administrator. Graded maximal exercise test (GXT) reports are exported in comma separated volume format and transferred in a similar fashion.

Intervention adherence is performed primarily through actigraphy and heart rate data recorded using the Fitbit platform. Each participant is provided a Fitbit device linked to a unique Fitbit account without any associated protected health information. Gmail accounts are readily created and managed by capitalizing on the Gmail feature of ignoring any email address text following a "+". Thus, unique Fitbit accounts can be created for all participants and managed through a single Gmail account; i.e. examplestudyemail+1@gmail.com and examplestudyemail+2@gmail.com can be the first and second Fitbit account user names for participants 1 and 2, but operate from a single Gmail account, examplestudyemail@gmail.com.

Fitbit data are accessed via the Fitbit Web application program interface (API) and a study-specific developer app. Unique Fitbit accounts are authorized to collect steps, activities, heart rate and sleep data. A collection of scripts (Fig 1) access data daily to transfer complete days from Fitbit servers to KU ADRC servers. Step, activity, and heart rate data are saved in comma separated volume format. A variety of fail-safe functions are embedded within these scripts to notify study staff and participants of potential problems. Harvested data are processed to count wear time, moderate intensity activity, and to summarize heart rate during activity before being stored in the working data directory. To corroborate Fitbit reports, participants report their weekly exercise using a REDCap survey.

Together, this system minimizes human touches to the data and almost eliminates the transfer of data from paper source documents to electronic data captures. We estimate that this automated transfer process saves study staff at least an hour per week. Eleven of 26 KU ADRC staff reported transferring data from a source document to an EDC as one of their responsibilities for a median time of 60 minutes per week (Q1:60 min, Q3: 120 min).

## Data curation

Once the source data enter the working dataset, they go through a curation phase in which data are read, quality checked, aggregated, and summarized to produce project-level '.Rdata' files for consistent reference during output. Fig 1 shows the general data and script flow.

The data curation process is unique by source type (i.e. DXA, GXT, cognitive testing, etc.). For sources that require more complex interpretation, i.e. ePrime cognitive tests, this process is done in its own script. For relatively easily interpretable sources, i.e. REDCap, this process is a subset of a larger script. However, each data curation process has a similar approach. 1) Find all source documents from an overarching directory. 2) Read source document. 3) Quality check data by ensuring reliable identifiers, dates, time points, and appropriate data types. 4) Aggregate all clean data from an individual data source into a single data frame. 5) Summarize data by merging when appropriate and creating project-wide variables that will be used in downstream output data, i.e., participant-specific week of intervention, heart rate range, etc. This step can include extensive data analysis, as in the case of calculating exercise adherence

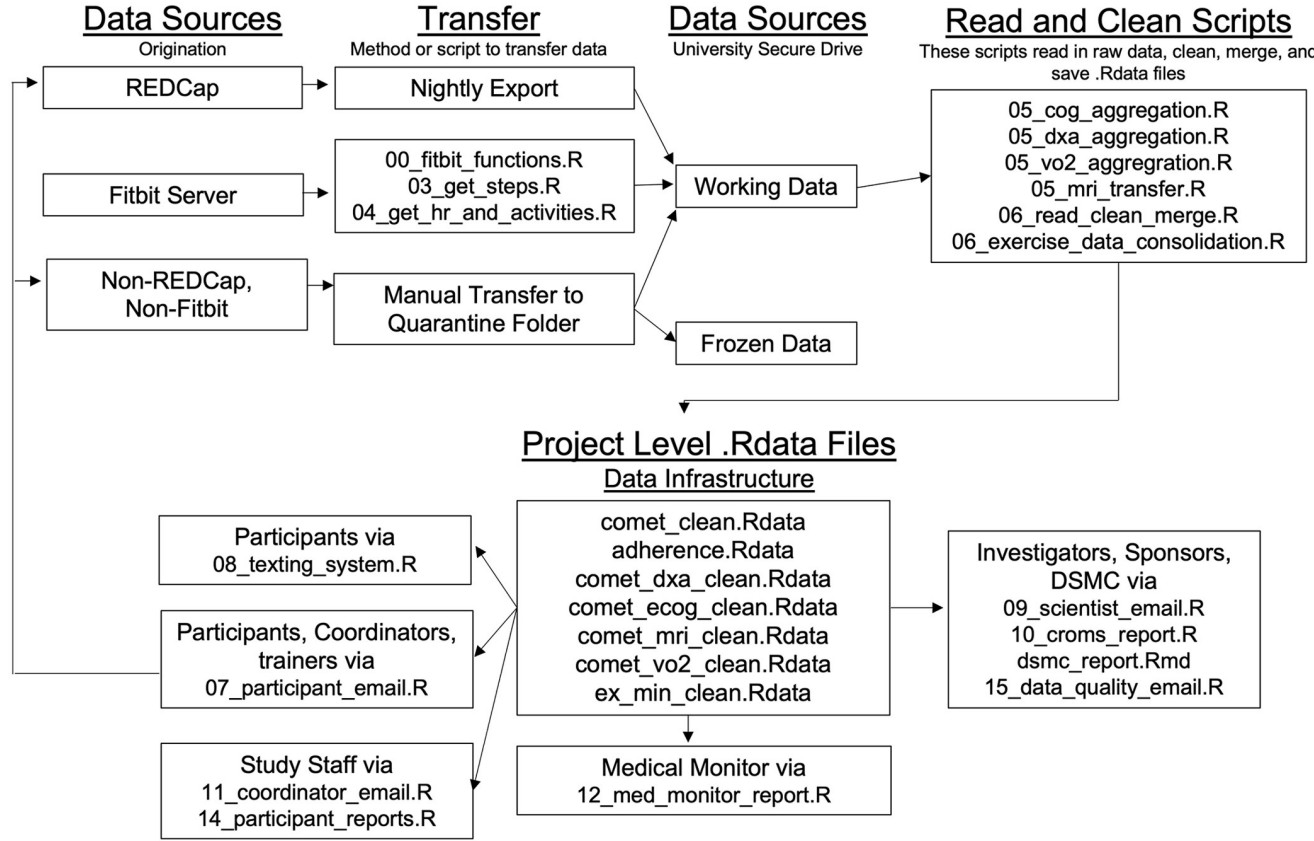

**Fig 1. Data flow.**

when multiple clean exercise sources are joined, go through additional cleaning, and adherence is estimated through an algorithm. 6) Save the dataframe in a '.Rdata' file in a clean directory for all downstream reference.

Regardless of complexity, the end product of data curation is a curated, reproducible '.Rdata' file saved in a separate directory. This final dataset is the basis for all reporting and data monitoring. Downstream output to stakeholders refers to these reliably curated data and never to raw data. This data curation process reduces redundancies and therefore errors. Data are curated nightly. Each morning, data reflecting the previous day are available for interrogation, reporting, and review by appropriate parties. At no time do the principal investigators or raters have access to this summary dataset, or the component source documents. Any demographics and adherence metrics approved for PI and study team review are presented in reports and dashboards approved by the study statistician thus maintaining tight blinding.

## Stakeholders

A number of stakeholders, each with a different role, contribute to the success of the COMET study (Fig 2).

The COMET study team has identified and created several automated reports to direct information to stakeholders and facilitate communication between stakeholders. All automated reports use a similar set of processes dictated by university limitations, i.e. packages available, unsupported languages (Java), etc. Each process varies depending on the need, but

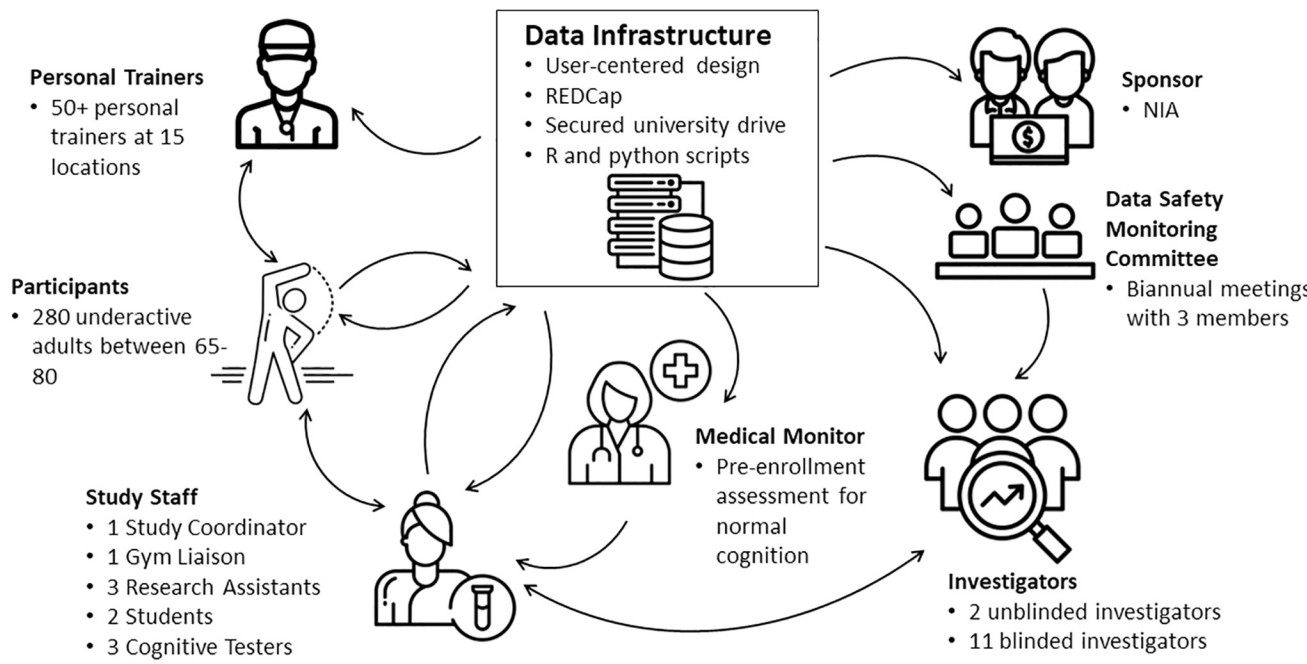

**Fig 2. Stakeholder information flow.**

the underlying structure tends to be similar. 1) Read in all relevant curated '.Rdata' files. 2) Produce an output, usually an R Markdown document (word, PDF, or html). 3) Send the document to relevant stakeholders via a Python email module. The following describes these data access reports for each stakeholder group.

**Participants.** Timely and usable information flow to and from participants is essential to a successful study. Participants must receive easily interpretable information at the appropriate time, i.e., exercise prescription, missing data, adherence feedback, and potential problems. Similarly, study staff must receive information from participants as quickly as possible, i.e. injuries, illnesses, potential problems, etc. To develop a dialogue that meets these needs for a large group of participants with diverse communication and technology preferences, and a busy study staff, we developed a series of touch points across many modes of communication with an emphasis on minimal study staff time burden and minimal barriers to response for participants. The starting point was weekly access to all essential study information. Participants, trainers, and study staff are copied on a participant's weekly email, delivered in html in the body of the email, which includes their exercise prescription, intervention performance feedback, upcoming testing dates, troubleshooting guide, and missing data. This setup allows all relevant stakeholders to have access to the same information, minimizes barriers to use, minimizes barriers to conversation, and provides standardized adherence feedback which supports sustained engagement, goal achievement, and sense of accomplishment. Thirteen of 26 KU ADRC staff reported emailing participants about study activities as one of their responsibilities, with a median time of 10 minutes per email (Q1:5 min, Q3: 10 min). Throughout the COMET study alone, we anticipate more than 15,400 automated emails over 237 weeks of conducting the study. We estimate saving study staff 650 minutes per week by automating these extended participant emails.

For timely information, automated messages are generated to trigger reminders for upcoming appointments, Fitbit charging and syncing, extended absences, and other problems.

Participants have the choice of receiving these via email or text, with the option to substitute phone calls as preferred. In the first year of operation, we sent 2301 automated text messages. Extrapolating across the expected 237 weeks of the study, we expect to send over 10,487 text messages. 6 KU ADRC staff reported texting participants about study activities as one of their responsibilities for a median time of 2 minutes per text (Q1:1 min, Q3: 4.5 min). Thus, we estimate saving study staff about 44 minutes per week.

In COMET, participants receive regular check-in calls and in-person visits at the gym, during which overlapping information is provided. The automated email and messaging reporting can be substituted for paper and phone-calls for those that prefer. Written and mailed communication can also be used. This system covers email, text-messaging, phone, in-person, and written communication thus providing our older adults with their choice of how to engage with study staff and reducing study staff time-burden by automating most outgoing information.

**Study staff.**   Clinical trials, especially those as complex as COMET, require a significant amount of person power and time from trained and skilled study staff (estimated 75 hours per week with a peak of 137 hours). At peak enrollment, we anticipant study staff will supervise 115 participants in the field across the many exercise sites noted above. Ready access to key metrics help study staff identify needs, maintain a task priority list, intervene with wayward participants, and monitor the trial successfully. In COMET, the coordinator and study staff use a collection of daily updated reports. In a centralized report, study staff are notified of database errors, potential protocol deviations, upcoming tasks, missing data, Fitbit issues, participant correspondence, intervention adherence, and adverse events. In individual participant reports, participant data from disparate sources, i.e., REDCap, Fitbit, VO2, etc., are aggregated into a single comprehensive report. These reports were designed with an iterative, user-centric process, with study staff to support study maintenance and reduce time burden. 14 of 26 KU ADRC staff reported identifying coordinator-related problems in call lists, scheduled appointments, database errors, missing data, adverse events, protocol deviations, and potential adherence issues as one of their responsibilities for a median time of 240 minutes per week (Q1:75 min, Q3: 345 min). We estimate saving study staff approximately 240 minutes per week by automating coordinator-related reporting.

**Interventionists.**   Community based interventionists, e.g. personal trainers, are key for the community exercise model deployed in our studies. These intervention supervisors receive human subjects research and protocol training, but are employees of their respective fitness facilities, and invoice their time to the study. Supporting these interventionists with participant specific information is key to effective intervention delivery. Interventionists educate participants, receive initial adverse event reports, and support exercise per protocol in a safe and effective manner. Facilitating communication and delivering consistent information between trainers, participants, and coordinators is essential for the successful delivery of the intervention. As such, all three stakeholders, trainers, participants, and coordinators, are included on the weekly emails.

**Medical monitor.**   Key to adjudication of both inclusion/exclusion criteria and adverse event evaluation is the involvement of our medical monitor. For COMET, the medical monitor reviews baseline cognitive performance for impairment concerns. Nightly, baseline demographic, medication, and cognitive data from several difference source are aggregated into a PDF report for each enrolling participant, which is sent electronically to the monitor for adjudication and signature via DocuSign.

**Investigators, sponsors, and data and safety monitoring.**   Safety and data monitoring responsibilities and rigor have increased significantly for principal investigators. COMET staff have developed a series of reports to automate these responsibilities and assist in decision

making for principal investigators. Weekly trial status reports are generated for the principal investigators to identify support needs, evaluate recruitment trends, and review adverse events. Five of 26 KU ADRC staff reported compiling enrollment, demographic, or other data to help make decisions about a study as one of their responsibilities for a median time of 15 minutes per report (Q1:10 min, Q3: 15 min); thus, we estimate saving study staff approximately 15 minutes per week. As data and safety monitoring committees (DSMC) are now required for most clinical trials, an automated DSMC report in the NIH template format, generated via Markdown as an easily editable Word document, is updated and delivered weekly to investigators along with the trial report. This eliminates the well-known phenomenon of last-minute scrambling by study coordinators to assist investigators in the generation of a report for DSMC meetings. Only 3 KU ADRC staff reported compiling DSMC reports, predominantly focused an AE reporting, for a median time of 90 minutes per report (Q1:75 min, Q3: 110 min). As COMET's automated DSMC reporting generates not only AE reporting, but also consort diagrams, screening and enrollment, missing data, adherence, and fidelity in both closed and open reports, we estimate that study staff save at minimum 90 minutes bi-annually, but likely much more.

The third and final investigator-focused report generates an interactive HTML report to evaluate data quality for missingness, time-drift, and tester differences. This report is delivered as separate open and closed reports to blinded and unblinded investigators respectively with the weekly trial report. Lastly, to satisfy sponsor requirements, nightly recruitment reports are generated to satisfy the new Clinical Research Operations and Management System, CROMS. Only 4 KU ADRC staff reported submitting CROMS or Research Performance Progress Report (RPPR) data for a median time of 630 minutes per report (Q1: 46 min, Q3: 916 min). Given that COMET's reporting requirements for CROMS are not as extensive as others in our center, we estimate automating CROMS reporting saves study staff approximately 46 minutes per month. Each of these reports was and continues to be designed iteratively with the stakeholder and the needs of the study.

## Discussion

In this manuscript, we have presented our development of a suite of scripts and data management procedures that increase our efficiency for conducting rigorous, randomized controlled trials with limited personnel across many intervention sites. With study locations in both metropolitan and rural areas, it is imperative that study recruitment, exercise protocol and data collection remain consistent. The automated nature in which COMET is being conducted achieves this goal. Data generated in this study is available in near real time for use by a variety of stakeholders to maximize study quality and reduce staff burden. Our data management protocol protects raw source documentation and maintains role-based blinding while providing key elements of intervention and operations data to participants, interventionists, study staff, and study investigators for trial oversight. We estimate that COMET's data infrastructure automates, at minimum, 17 hours of study staff responsibilities per week. It is worth noting that the time savings have allowed the principal investigators to encourage staff to spend less time doing route tasks like data entry and more time developing relationships with participants and improving data quality.

The COMET study team has a broad range of experience in clinical trials, exercise interventions, exercise psychology, and aging. The team has no professional coding experience, or advanced understanding of data management. Rather, the success of the COMET data management system is in our ability to deploy a useful and efficient system of data support despite our technical deficiencies. The individuals who built and maintain the REDCap project and

the R data curation system are hobbyist programmers with no formal training. Their extensive knowledge of research best practice, clinical trial conduct, and willingness to learn R were key to the success of this system. This underscores the utility and ease of use for open-source platforms of data management. We believe that investigative teams with a broad range of professional experience in data management can effectively build on open-source tools to improve operational efficiency [28].

We expect that the open-source nature of our system creates opportunities for broader application. For example, dietary interventions are another branch of behavioral investigation requiring significant communication and intensive monitoring. Adapting the system for dietary monitoring could reduce error in nutrient analyses. Additionally, incorporating data management similar to the approach used in COMET for diet interventions could enhance the interaction between investigators, dieticians, and participants who are often confused by the information provided to change diet. Shared recipes that calculate nutritional information, and correspondence that supports food logging that could calculate macronutrients and micronutrients in meals are options under this system. This system could also be applied to future multi-site studies with greater geographic diversity, allowing for a greater investigational reach. Text message is nearly universally available, even in rural communities, at low cost, making the flexible and multimodal correspondence particularly adaptable. Beyond research, the communications aspects of the system could be leveraged and adapted for community impact programs. The ability to communicate effectively and efficiently, motivate individuals and small groups of adults in need of lifestyle behavior changes and organization of professionals (health coaches, personal trainers, dieticians and more) may help create a more significant impact in rural and geographically isolated regions than current, traditional approaches [29].

As previously noted, the clinical trial enterprise would benefit from an increased effort to automate aspects data management [30]. We have detailed an extensive portfolio of scripting and data management that can be accomplished without extensive network or informatics expertise. We recognize that all these parts have been done and reported on previously. We do not suggest we've stumbled on any novel approaches. However, our innovation is to incrementally advance the idea of what can be achieved by the typical academic research study team. COMET has pulled together many disparate pieces and best practices into an entire system for semi-automated clinical trial data management and use. Emphasizing Open Science and FAIR Data principles [31] we have created a Git repository to share this work freely with the hope that our system will encourage other study teams to adapt and improve upon this system for clinical trial support: https://github.com/cometstudy/OSSforEfficientClinicalTrialSupportCOMET.

## Supporting information

**S1 File. Step-by-step protocol, also available on protocols.io.**
(PDF)

## Author Contributions

**Conceptualization:** Jonathan Clutton, Robert Neal Montgomery, Ashley R. Shaw, Amanda N. Szabo Reed, Eric D. Vidoni.

**Data curation:** Jonathan Clutton.

**Formal analysis:** Jonathan Clutton, Eric D. Vidoni.

**Funding acquisition:** Amanda N. Szabo Reed.

**Investigation:** Jonathan Clutton, Robert Neal Montgomery, Dinesh Pal Mudaranthakam, Amanda N. Szabo Reed, Eric D. Vidoni.

**Methodology:** Jonathan Clutton, Robert Neal Montgomery, Dinesh Pal Mudaranthakam, Erin M. Blocker, Ashley R. Shaw, Amanda N. Szabo Reed.

**Project administration:** Jonathan Clutton, Robert Neal Montgomery.

**Resources:** Jonathan Clutton.

**Software:** Jonathan Clutton, Robert Neal Montgomery, Dinesh Pal Mudaranthakam.

**Supervision:** Robert Neal Montgomery, Amanda N. Szabo Reed, Eric D. Vidoni.

**Validation:** Erin M. Blocker.

**Writing – original draft:** Jonathan Clutton, Robert Neal Montgomery, Dinesh Pal Mudaranthakam, Erin M. Blocker, Ashley R. Shaw, Amanda N. Szabo Reed, Eric D. Vidoni.

**Writing – review & editing:** Jonathan Clutton, Robert Neal Montgomery, Dinesh Pal Mudaranthakam, Erin M. Blocker, Ashley R. Shaw, Amanda N. Szabo Reed, Eric D. Vidoni.

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
